# Internal Quality Assessment of East African Highland Cooking Banana (*Musa* spp.) Flour: Significance for Breeding and Industrial Applications

**DOI:** 10.3390/foods12234323

**Published:** 2023-11-29

**Authors:** Elizabeth Khakasa, Charles Muyanja, Robert Mugabi, Yusuf Mukasa, Mary P. Babirye, Brian Balikoowa, Priver Namanya, Jerome Kubiriba, Ivan K. Arinaitwe, Kephas Nowakunda

**Affiliations:** 1National Agricultural Research Organisation (NARO), National Agricultural Research Laboratories (NARL), Kampala P.O. Box 7065, Uganda; lizkhakasa@gmail.com (E.K.); mukasayusuf89@gmail.com (Y.M.); babiryeprudencemary12@gmail.com (M.P.B.); balikowabrian787@gmail.com (B.B.); bwesigyep@gmail.com (P.N.); jkubiriba2012@gmail.com (J.K.); ivankabiita@gmail.com (I.K.A.); nowakunda@gmail.com (K.N.); 2Department of Food Technology and Nutrition, School of Food Technology, Nutrition and Bio-Engineering, Makerere University, Kampala P.O. Box 7062, Uganda; ckmuyanja@gmail.com

**Keywords:** cooking banana flour, functional properties, pasting properties, banana breeding, industrial application

## Abstract

This study assessed the internal quality traits of East African Highland cooking banana flours, exploring their significance for breeding and potential industrial applications. Twenty cultivars (nine hybrids and eleven landraces) were used. Swelling power capacity, water solubility, water absorption capacity, water absorption index, freeze–thawing stability, and pasting characteristics of banana flour were assessed using standard methods. The results showed that cultivars with high swelling power also exhibited a high water absorption capacity and water absorption index, thus making them suitable for bakery industries. The water absorption capacity ranged between 5.66% (N2) and 11.68% (N11). Landraces KBZ (9.01) and NKYK (8.05), and hybrids N11 (11.68) and N9 (8.48) are suitable as thickeners due to high WAC. Hybrids (N7, 27.83%, and N9, 22.59%) and landraces (NMZ, 32.69%, and NFK, 34.24%) had low freeze–thawing stability, hence it is applicable as a food stabilizer. Landrace NKT (19.14%) and hybrid N9 (16.95%) had the highest solubility, and landrace KBZ (6.93%) and hybrid N3 (6.66%) had the lowest solubility. Landraces MSK (6265), NKY (3980), and NFK (3957), and hybrids N6 (3608), N7 (3505), and N9 (3281 RVU) had high peak viscosity. The trough viscosity, final viscosity, and breakdown viscosity of cultivars varied from 422.5 to 5004 RVU. The landraces MSK (5021 RVU) and NFK (4111 RVU) had the highest final viscosity, making them suitable for application in the food industry for thick and stable sauces. Landrace TRZ had the lowest pasting temperature (62.7 °C), making it advantageous for use where fast gelatinization is required, hence saving energy costs and cooking time. These findings suggest that the genetic attributes inherent in cultivars can be incorporated into breeding programs targeting required traits for industrial application.

## 1. Introduction

Bananas are among the most important crops grown globally [1]. They are classified into four main categories based on their primary culinary use and taste characteristics: juice or beer bananas, dessert bananas, roasting bananas, and cooking bananas. Cooking bananas are further classified into two categories; the first category is consumed only after it ripens. It is prepared by either boiling, frying, or steaming, and they are commonly known as plantains. The second type is consumed at its green stage, but only after cooking, and this can either be boiled or steamed. These comprise mainly the East African Highland Bananas (*Musa* AAA) [1], and are characterized by being starchy. The East African Highland Cooking Banana, also known as ‘Matooke’ in Uganda, is a staple in the Great Lakes region of Africa, feeding more than 50 million people [2,3,4]. It is also widely traded within countries, and has great industrial potential [5].

Unfortunately, the local banana cultivars are very susceptible to diseases and pests, necessitating the development of hybrids with disease/pest resistance and drought tolerance, among other traits [6,7]. Some of the improved genotypes are not accepted by end users due to their lack of acceptable sensory characteristics, which is a big problem in banana breeding [8,9].

The quality of cooking bananas depends on the intrinsic properties that contribute to acceptability. Variations in intrinsic properties of cooking bananas have an effect on the quality of the food and banana-based products. A number of studies have reported the properties of banana flour [10,11].

In a breeding program, it is necessary to test the quality of materials in the early stages of breeding to eliminate populations or breeding lines that do not qualify for advancement. Screening based on only pests, disease resistance, and agronomic traits has resulted in low uptake of improved bananas [12]. The breeding process is an expensive, slow, and time-consuming procedure that involves the selection of genotypes at different stages of the breeding cycle for advancement to the next [13]. Breeding, therefore, has to incorporate the internal properties that contribute to sensory quality, alongside the yield and resistance potentials to pests and diseases. Studies that relate functional and pasting properties of food to breeding exist for other crops [14,15]; however, there is no literature for such studies in cooking banana, and this work finds that an area of novelty, hence its foundation.

Being a versatile fruit, banana has not only been a dietary cornerstone, but also a cultural emblem, deeply embedded in the culinary traditions of the region. Beyond its traditional use as a staple food, recent innovations have unveiled a promising avenue—the transformation of these bananas into flour, opening doors to a diverse array of culinary applications [16,17].

This study embarks on a thorough exploration of this transformative process, with a keen focus on the intrinsic qualities that define the resulting banana flour. Beyond the surface, we delve into the aspects of functional properties and other critical attributes, such as the pasting characteristics that underpin its quality. This endeavor carries profound implications for both banana breeding programs and the burgeoning industrial sector. By dissecting the internal characteristics of banana flour, we aim to provide an understanding that transcends traditional notions of food quality. This knowledge forms the bedrock upon which future breeding efforts can be built, with the ultimate goal of producing banana varieties that excel not only in yield but also in nutritional content and culinary versatility. Simultaneously, our insights hold promise for industries seeking innovative, sustainable ingredients, potentially catalyzing a paradigm shift in the utilization of banana flour.

As we embark on this journey through the heart of East African Highland Cooking Bananas, we anticipate that our findings will not only enrich the scientific discourse, but will also have a tangible impact on the livelihoods of communities in the region, heralding a new era of agricultural and industrial possibilities.

The banana breeding program is challenged with the absence of a tool that can screen large numbers of genotypes in a short time, but is also able to assess fruit quality to aid in the screening and selection of elite material during the early stages of breeding. As a result, time and resources are wasted to develop acceptable hybrids, hence the need to investigate the intrinsic components that contribute to quality, and can support the breeding of highly acceptable cooking bananas.

Most studies put their focus on the functional properties of bananas [10,18,19,20,21,22,23,24,25]. No study has integrated the use of landraces and hybrids in the evaluation of functional properties of cooking banana to aid selection in breeding. The main objective of this study was to conduct an internal quality assessment of East African Highland Cooking Banana flour (*Musa* spp.) in order to explain its intrinsic attributes, with a particular emphasis on its significance for breeding programs and potential applications in the industrial sector.

Specifically, this study aimed to: (i) examine the functional properties of banana flour to understand its suitability for various food processing applications; (ii) offer practical recommendations for leveraging the findings in breeding efforts and industrial applications.

## 2. Materials and Methods

### 2.1. Materials

#### Cooking Bananas and Preparation of Flour

Twenty cooking banana varieties, nine hybrids and eleven landraces, grown in the fields of the National Agricultural Research Laboratories (NARL) and the International Institute of Tropical Agriculture (IITA) fields in Sendusu in central Uganda were used in this study. The criteria for selection were based on the possession of very good, medium, and very bad sensory quality traits obtained through routine indicative sensory tests conducted at the Food Biosciences Laboratories, Kawanda. The landrace genotypes included Kazirakwe, Kibuzi, Kisansa, Namwezi, Nakinyika, Nakitembe, Mpologoma, Nfuuka, Tereza, Mbwazirume, and Musakala, and the hybrids were NARITA 1, NARITA 2, NARITA 3, NARITA 6, NARITA 7, NARITA 8, NARITA 9, NARITA 11, and NARITA 12, all harvested over a period of six years (Table 1). The NARITAs are cooking banana hybrids developed by the National Agricultural Research Organisation (NARO) and IITA [26]. While some cultivars, such Nakitembe, Musakala, and Nfuuka, belong to specific clone sets, hybrids are not assigned to any specific clone set, and are denoted in this column by NA (not applicable).

Fruits were obtained from the second and third hands of each bunch, and randomly mixed before selecting those that were peeled. The peeled fruits were sliced and dried in a Gallenkamp oven (Gallenkamp, Model 300, Cambridge, UK) at 65 °C for 24 h by a slightly modified method of Dadzie and Orchard [27]. The dried banana slices were ground in a mortar to obtain flour, which was used for further analysis.

### 2.2. Methods

#### 2.2.1. Functional Properties

##### Swelling Power Capacity

A modified method for the determination of swelling power was used, as described by Yu et al. [28]. With the help of a Mettler Toledo weighing balance (Mettler Toledo, Model MS 204S/01, Greifensee, Switzerland), the dried banana flour sample (0.35 g) was weighed into falcon tubes of 50 mL, and 10 mL of distilled water was added. The samples were heated in a shaking water bath (Grant Instruments, Model OLS 200, Cambridge, UK) at 80 °C for 30 min. The starch solution was cooled immediately to room temperature, and then centrifuged (Centurion Scientific, C2 Series, Model Pro-Sep E, West Sussex, UK) at 2050 × 10 rpm for 15 min. The supernatant was removed and the sediment was weighed. Swelling power was calculated from the formula
(1)$$\mathrm{Swelling power}\;(g/g)=\frac{\mathrm{Weight of wet sediment}}{\mathrm{Weight of sample}\;(\mathrm{dry weight basis})-\mathrm{Weight of dried supernatant}}$$

##### Water Solubility

The method described by Marta et al. [29], with slight modifications, was used in the determination of water solubility. The dried banana flour sample (0.35 g) was weighed into falcon tubes, and 12.5 mL of distilled water was added. The sample was heated in a water bath at 65 °C for 30 min, followed by centrifugation at 1800 × 10 rpm for 15 min. The supernatant was removed and poured into a preweighed crucible, and the solubility was evaluated in a constant-temperature oven. Solubility was computed as follows
(2)$$\%Solubility=\frac{\mathrm{Weight of dried supernatant}\;(g)}{Weight\;of\;Sample\;(dry\;weight\;basis)(g)}\times 100$$

##### Water Absorption Capacity (WAC)

A procedure by Torruco-Uco and Betancur-Ancona [30] was used to determine water absorption capacity. A 1 g sample of banana flour was weighed in a 15 mL centrifuge tube, and 10 mL of distilled water was added. The sample was agitated on a vortex mixer for 2 min before centrifuging at 1500 × 10 rpm for 35 min. The supernatant was separated and its volume measured before being discarded. Water absorption capacity was calculated from the formula
(3)$$WAC\;(g/g)=\frac{Volume\;of\;water\;absorbed}{Weight\;of\;sample\;(dry\;basis)}$$

##### Water Absorption Index (WAI)

Water absorption index (WAI) was determined according to a method described by Kraithong et al. [31], and is calculated from the formula
(4)$$WAI=\frac{Weight\;of\;wet\;sediment\;(g)}{Dry\;weight\;of\;flour\;(g)}$$

##### Freeze–Thawing Stability (% Syneresis)

Freeze–thaw stability was determined using a modified method of Marta et al. [29]. Distilled water (100 mL) was added to 5 g banana flour. The mixture was heated in a water bath at 95 °C for 30 min. Starch paste (10 g) was placed in a centrifuge tube and the sample was frozen at −20 °C for 48 h. The frozen starch was thawed at room temperature, followed by centrifugation at 2050 × 10 rpm for 15 min. The separated water was weighed, and the syneresis was calculated from the formula;
(5)$$Syneresis\;(\%)=\frac{Weight\;of\;separated\;water}{Weight\;of\;starch\;paste}\times 100\%$$

#### 2.2.2. Pasting Properties

The behavior of the starch granules (pasting properties) of the matooke flour was determined using a rapid visco analyzer (Model 3D, Newport Scientific Pty Ltd., Sydney, Australia) following a method described by Du et al. [32]. A 2.5 g cooking banana flour sample was weighed into a canister, followed by 25 mL of distilled water. This was then inserted into the rapid visco analyzer (RVA). The sample was mixed by a paddle, with rotation set at 160 rpm during the measurements, except for 960 rpm during the first 10 s. Each matooke flour suspension was equilibrated at 50 °C for 1 min, heated at a rate of 6 °C/min to 95 °C, and then kept at 95 °C for 5 min. This was followed by cooling the suspensions to 50 °C at a rate of 6 °C/min.

#### 2.2.3. Statistical Analysis

Data were analyzed using XLSTAT software version 2022.3.1. The Shapiro–Wilk test was used to test for normality of the data since the sample size was small (*n* = 60). Pearson’s correlation analysis was conducted to observe existing relationships between the variables, and linear regression to determine the goodness of fit of the variables.

## 3. Results and Discussion

### 3.1. Functional Properties

#### 3.1.1. Swelling Power (SP)

Swelling power (SP) is defined as the ability of starch granules to absorb water during the heating process, which causes them to increase in size [33]. SP ranged between 7.82 (N2) to 12.55 g/g (N11). SP of the hybrids varied from 8.61 to 12.55 g/g, whereas landraces varied from 8.88 to 10.48 g/g (Table 2). A high swelling power (SP) shows that the cooking banana flour has a high potential to absorb and retain water (Figure 1), which indicates a higher gelatinization capacity [34]. The formation of gels is an important factor in various food applications because of its importance in texture, stability, and viscosity. KBZ, a landrace cultivar, had the highest SP, which implies that its potential to form gels is high, as compared to the hybrid cultivars. High SP is an important characteristic in food manufacturing, with applications in thickening agents, soups, fillings, and production of gels.

These results are in agreement with those reported by other authors [33,35,36,37,38], who noted that SP relates to pasting characteristics and rheological properties. On the contrary, Alimi et al. [34] reported a range of 1.75–3.2 g/g on false banana, whereas Olatunde et al. [11] reported values of 1.38 and 2.14 g/g for native banana, which is quite lower than the results obtained in this study. This implies that landraces have starch structures that are more open compared to hybrids, which makes them absorb more water during processing, hence making them suitable for use as adhesives in the food and nonfood industries. The greater SP values of N11, N9, N8, and N7 in the hybrids, and MPO and TRZ in the landraces, suggest a high gelatinization capacity. Low swelling power is not a good attribute for application in the food industry, as it affects the final yield of the products.

The values for the swelling power recorded in this study were higher than those reported by Akubor et al. [39]. This could be attributed to the intrinsic differences in the cultivars, but also the starch modification methods applied. Akubor et al. [39] employed pregelatinization and annealing.

Swelling power has been reported to increase with increasing temperature [37]. Additionally, it is also affected by chemical composition, structural characteristics, and granule particle size [40]. Pearson’s correlation coefficients showed that SP was positively correlated with water absorption index (*r* = 0.97) and water absorption capacity (*r* = 0.60). This means that cultivars that had a high SP were able to absorb more water as a result of high swelling capacity of the granules. Swelling is a very good parameter for the bakery industry, as it increases the volume of the dough produced, and hence a greater yield (number of loaves at the end of mixing).

#### 3.1.2. Water Absorption Index (WAI)

Water absorption index (WAI) of the cooking banana flours varied between 7.16 (N2) and 11.16 g/g (N11). Water absorption index determines the volume occupied by the starch granules after they swell in excess water, and indicates the integrity of starch in aqueous dispersions [32]. WAI is described by the interactions that take place between water molecules and banana flour. High WAI promotes high moisture retention in food, and may be useful in the bakery industry for cakes and bread that need to be kept moist, whereas low WAI may be useful for products that require crunchiness as an attribute.

Pearson’s correlation coefficients showed that WAI was strongly and positively correlated with SP (*r* = 0.97) and WAC (*r* = 0.62). This suggests that cultivars with high SP had the ability to take in more water, hence an increase in the water absorption and index capacities. This implies that the landrace cultivar MPO, and hybrid cultivars N7, N8, and N11, can be used in the food industry as thickeners because of their ability to absorb large amount of water and form gels (Figure 1).

#### 3.1.3. Water Absorption Capacity (WAC)

The results showed that the water absorption capacity (WAC) for the cultivars ranged between 5.66 (N2) and 11.68% (N11). Similar results have been reported by other authors [20,41]. The WAC of hybrid cultivars ranged from 5.66 to 11.68, and that of landraces from 6.55 to 9.01. This implies that both hybrid and landrace cultivars that had a high WAC (N9, N11, KBZ, and NKYK) are suitable for use in food and pharmaceutical industries. Only one hybrid cultivar (N2) showed a very low WAC, and hence was not suitable for use in the food industry. Figure 1 shows a graphical representation of water absorption capacity of the selected banana flours. As the WAC increased, WAI and SP also increased. Pearson’s correlation coefficient showed that WAC was positively correlated with WAI (*r* = 0.62) and swelling power (*r* = 0.60), but negatively correlated with freeze–thaw stability (*r* = −0.40).

WAC of flour plays a big role in the process of food preparation by allowing a high amount of water to be taken in by the starch molecules, which aids mixing to obtain a desired consistency and influencing sensory properties [32]. On the other hand, higher values for WAC (89.33%) were reported for false banana [34]. Additionally, WAC values for native banana (136.5%) were also noted to be high [11], as well as for hybrid plantain (86.5 to 88%) cultivars [36].

#### 3.1.4. Freeze–Thawing Stability (Syneresis)

Freeze–thawing stability (syneresis) for the cultivars ranged between 22.59 and 70.53%. As shown in Figure 2, cultivar N3 had the highest syneresis, whereas cultivar N9 had the lowest. Similar results are reported by other authors [29,42]. Syneresis is the removal of water from frozen gels. Low values are indicative of good stability of the starches, as they are less likely to downgrade [20]. High syneresis values, however, indicate low stability of the starch, rendering it not fit for use in refrigerated foods as a gel or thickener [41]. Pearson’s correlation coefficients showed inverse relationships between syneresis and SP (*r* = −0.49), WAI (*r* = −0.49), and WAC (*r* = −0.40). The hybrid cultivars that exhibited low syneresis values are N9 (22.59%) and N7 (27.83%), whereas the landrace cultivars were NMZ (32.69%) and NFK (34.24%). These results are suggestive of possible application of these starches as food stabilizers in the food industry for the manufacture of sauces.

#### 3.1.5. Solubility

Results from this study showed that solubility ranged from 6.66 (N3) to 19.14% “(NKT)” “(Table 2)”. Solubility of hybrids ranged from 6.66 to 16.95%, whereas that of landraces varied from 6.13 to 19.14%. Landraces exhibited much higher solubility compared to hybrid cultivars (Figure 2). A very weak positive correlation between solubility and WAC (*r* = 0.12) was observed. Solubility defines the extent to which starch chains interact within crystalline and amorphous domains, influenced by the amylose:amylopectin ratio and characteristics [43,44]. It is also defined as the property of the flour to dissolve in a solvent. A number of authors reported similar findings [29,33,38,41,44]. High solubility of flours is an indicator for high digestibility [45]. Solubility was highest for NKT (19.14%) and lowest for N3 (6.66%). From this study, the landraces NKYK, MPO, MSK, TRZ, and NKT had high solubility values together with the hybrids N9 and N11, and therefore can be selected by breeders for use in industry.

### 3.2. Pasting Properties

The RVA profile of cooking bananas is shown in Table 3. Significant variations (*p* < 0.05) in the pasting properties were observed across the different cooking banana cultivars.

#### 3.2.1. Peak Viscosity (PV)

The average peak viscosity (PV) of the cultivars varied from 457 in KIS to 6265 RVU in MSK. The highest peak viscosity (PV) was observed in the landrace cultivars MSK, NFK, and NKY, and hybrid cultivars N6, N9, and N7. High PV is suggestive of the increased ability of flour from these cultivars to absorb and bind more water through its hydrogen bonds [31]. Peak viscosity, which is indicative of increased ability to form pastes when cooked, shows the strength of the paste that is formed during the gelatinization process [46]. Similar results were reported by several authors [20,22,29,33]. The results from this study, however, were higher than those reported by other authors for banana [11,19,39]. The differences in the results could be due to starch being subjected to modification processes such as pregelatinization, acetylation, and oxidation. The authors noted that modification significantly reduced (*p* < 0.05) the peak, final, trough, and setback viscosities of the bananas. However, these authors’ results contradict what was reported by Akubor et al. [39], that pregelatinization and annealing increases banana starch pasting properties.

Peak viscosity is the measurement of how a starch granule freely swells when it takes in water, and this is achieved when the pasting temperature is reached. These results correlated positively with the swelling power, as similarly reported by Akubor et al. [39]. High PV is an indicator of large granular sizes of the starch molecules. Previous authors have reported that peak viscosity is related to water absorption capacity; the higher the peak value, the more the water is absorbed, an indication of high starch content [29,31]. The results of this study, however, showed inverse relationships between peak viscosity and water absorption capacity (*r* = −0.12), although it is weak. The results suggest that cultivars with high PV (MSK, NKYK, NFK, N6, N9, and N7) can be used as parents during breeding programs targeted at increased starch content.

The implications arising from the peak viscosity findings in relation to breeding indicate that the landrace cultivar MSK exhibits the highest peak viscosity, thereby suggesting a comparatively higher starch content, or a greater propensity to form a viscous paste in comparison to other cultivars. Conversely, the hybrid KIS demonstrates the lowest peak viscosity, implying a lower starch content, or a diminished capacity to generate a thick paste.

When undertaking the breeding of cooking banana varieties, peak viscosity assumes significance as a pivotal parameter for the selection of desirable traits. Breeders pursuing the development of varieties with elevated starch content or enhanced thickening characteristics may accord priority to cultivars showcasing higher peak viscosity values, such as the landrace MSK. By incorporating the genetic attributes inherent in cultivars with high peak viscosity into breeding programs, it becomes feasible to augment the starch content and functional attributes of forthcoming banana varieties.

Equally, breeders aiming to satisfy specific applications where reduced viscosity or diminished thickening properties are preferable may consider cultivars exhibiting lower peak viscosity values, such as the landrace KIS. A comprehensive understanding of the variations in peak viscosity among distinct cultivars equips breeders with invaluable insights for the selection of parental lines, and the design of breeding strategies aimed at producing cooking bananas possessing desired functional characteristics.

#### 3.2.2. Trough Viscosity (TV)

Trough viscosity (TV), or the holding strength, is a measure of the resistance to flow or the thickness of a substance [46]. Results showed that trough viscosity of cooking banana varied from 422.5 to 4254 RVU. Trough viscosity was highest in cultivar KIS and lowest in MSK. Landrace cultivars MSK, NKYK, NFK, and NMZ, with hybrid cultivars N9, N6, N7, and N12 showed high TV values, whereas cultivars N2, KIS, and KBZ showed low TV values. Lower values of 272 to 640 RVU and 270.13 RVU have also been reported by Da Mota et al. [19] and Olatunde et al. [11], respectively. These results are in agreement with those of previous studies [20,22,34]. Pearson’s correlation coefficients showed that TV was related to PV (*r* = 0.95), FV (*r* = 0.82), and BDV (*r* = 0.51), as shown in Table 4. High TV is an indicator of high holding strength during processing, which implies that use of cooking banana flour with high TV could increase the holding strength of products during high heat treatments.

Trough viscosity is an important parameter related to the texture and mouthfeel of banana products, such as purees and processed foods. The higher the trough viscosity, the thicker and more viscous the banana pulp. Varieties with higher trough viscosity, such as the landrace cultivar MSK, may be suitable for applications where a thicker and creamier texture is desired. Banana breeding programs can target varieties with specific trough viscosity ranges to meet the requirements of different processing methods. For instance, bananas with lower trough viscosity, such as KBZ, KIS, and N2, may be preferred for applications such as juice production or smoothies, where a more fluid consistency is desired.

#### 3.2.3. Breakdown Viscosity (BDV)

The breakdown viscosity (BDV) values range from 414 to 2396 RVU. The landrace cultivar NFK exhibits the highest breakdown viscosity, indicating that it undergoes a significant reduction in viscosity during the testing period. On the other hand, the landrace KIS has the lowest breakdown viscosity, suggesting that it maintains a relatively higher viscosity compared to other cultivars. BDV is the difference between the peak and trough viscosities, and is described as the extent of the degree of disintegration of the granules or stability of the paste [47]. These results are in agreement with those reported by other authors [20,22,29]. The results from Pearson’s correlation showed relationships between BDV and PV (*r* = 0.78), FV (*r* = 0.71), TV (*r* = 0.60), and SB (*r* = 0.60). The landrace cultivars NFK, NKYK, and MSK, and hybrids N6 and N7 showed high BDV values, whereas cultivars KIS, KBZ, N2, and N11 had the lowest BDV. Low BDV is an indication for starch stability during the heating and stirring process, and flour from these cultivars would be recommended for use in the making of sauces and soups.

These results are valuable for banana breeding and the selection of new hybrids because breakdown viscosity is associated with several important characteristics, such as fruit firmness, texture, and shelf life. High breakdown viscosity generally indicates a softer and more easily digestible fruit, which may be preferred for certain consumer preferences. Low breakdown viscosity, on the other hand, may suggest a firmer and less mushy texture, which could be desirable for other purposes, such as processing or transport.

Through understanding the breakdown viscosity values of different landrace cultivars and hybrids, banana breeders can make informed decisions about which varieties to select for specific breeding objectives. For example, if the goal is to develop bananas with a longer shelf life, breeders may focus on cultivars with higher breakdown viscosity values, as they tend to have a firmer texture that can withstand longer storage periods. Likewise, if the aim is to create bananas with a softer texture, hybrids with lower breakdown viscosity may be preferred.

#### 3.2.4. Final Viscosity (FV)

The final viscosity varied from 567 to 5004 RVU. The landrace cultivar MSK showed the highest final viscosity, while the landrace KIS had the lowest final viscosity. The results are in agreement with those reported by other authors [20], but differ slightly from those reported by Olatunde et al. [11], as 399.8 RVU. Ahmed et al. [22] reported higher FV for the two cultivars studied. ‘Williams’ had an FV of 2960 RVU, and ‘Baradika’ had an FV of 3784 RVU. The high FV values were attributed to the high starch content of the varieties. The final viscosity is the parameter that defines the quality of a product because it is the point at which the cooked paste becomes stable, and this happens after it has been cooked and cooled [22]. High FV of the varieties increases their potential for use in the food industry as a thickener in the making of sauces [22].

The landrace cultivars MSK, NFK, TRZ, NMZ, and NKYK, and the hybrid cultivars N3, N12, N1, N6, and N7 were observed to have high FV, and therefore exhibit a great potential to be used as thickeners during food processing. The landrace cultivar MSK, having the highest final viscosity, suggests that it could be a potential candidate for breeding programs focused on improving viscosity-related characteristics. Breeders may consider using MSK as a parent or incorporating its genetic traits into breeding lines to enhance the final viscosity of new hybrids. In so doing, the development of bananas with thicker and more desirable textures is possible, which can be beneficial for various applications such as thickeners in sauces.

On the other hand, the hybrid KIS, displaying the lowest final viscosity, indicates that it may not be suitable for applications that require higher viscosity, as it may result in less creamy or thinner products. However, this hybrid’s lower viscosity could be advantageous in certain contexts. For instance, it could be preferable for beverages where a more liquid consistency is desired, such as juices or shakes. Moreover, if the goal is to develop banana hybrids with a range of viscosities to cater to different product requirements, KIS could be valuable as a parent to introduce lower viscosity traits into breeding programs.

#### 3.2.5. Setback Viscosity (SB)

The landrace cultivar NFK recorded the highest setback viscosity (2549 RVU), and KIS the lowest (525 RVU). Hybrid cultivars had SB values varying from 786 to 2266 RVU, whereas landraces ranged between 567 and 5004 RVU. Similar results have been reported by other authors [20,22,29]. On the contrary, SB results from this study are higher than those reported by Olatunde et al. [11], of 129.67 RVU; the difference could be attributed to the starch modification process that Olatunde et al. [11] used. SB was correlated with BDV (*r* = 0.57) and FV (*r* = 0.62). High SB viscosity values of cooking banana starches suggest a reduced retrogradation tendency, hence possible application in bakery products to minimize staling [11,22]. During the cooling process, starch molecules tend to reassociate, which is called setback. The setback viscosity is defined as the last phase of the pasting curve. It is also called the cooling stage. This is where the viscosity increases again as granules are cooled, and retrogradation also takes place [48].

Higher setback viscosity values, as observed in the landrace cultivar NFK, suggest a thicker and more stable gel formation after shear stress. This characteristic may be desirable in certain food products that require a more firm and stable texture, such as fillings or gels for bakery goods. On the other hand, the landrace KIS, with the lowest setback viscosity, may indicate a more fluid and less stable gel formation. This characteristic could be advantageous for applications such as sauces or products where a thinner consistency is desired.

Knowledge of setback viscosity values of different cultivars can help breeders select parent plants with specific viscosity characteristics to incorporate into their hybridization efforts. For example, if the goal is to develop a hybrid with a thicker and more stable gel formation, breeders may choose to cross the NFK landrace cultivar with other cultivars exhibiting desirable traits in terms of taste, disease resistance, or other agronomic characteristics. On the other hand, if a thinner consistency is desired, the KIS landrace or cultivars with similar viscosity properties may be selected as one of the parents in the breeding program.

Through the strategic selection of parent plants based on setback viscosity values, breeders can introduce and combine desired traits, leading to the development of new hybrids with customized viscosity profiles that align with specific food product requirements or consumer preferences.

#### 3.2.6. Peak Time

Peak time varied from 4.65 to 6.00 min, and similar results were reported by other authors [11,19,22]. Peak time is defined as the time at which the peak viscosity is achieved. The cultivars exhibited variations in their response or behavior, leading to differences in peak time. Peak time was inversely correlated with BDV (*r* = −0.40). The higher the peak time, the higher the resistance of the starch granules to break during the process of heating. This implies that cultivars that had a high peak time (KIS, NMZ, N1, N12, N3, TRZ, and MSK) also had an increased resistance to starch granule swelling during the process of heating. The hybrid cultivar KIS exhibited the highest peak time, while cultivar N2 displayed the lowest. Cultivars with shorter peak times, such as N2, may be preferred in situations where fast processing is required. For example, in making food products that need to be prepared quickly, or if there is a high demand for a certain product, a cultivar with a low peak time could be more suitable. This could be beneficial for industries that prioritize efficiency and production speed.

#### 3.2.7. Pasting Temperature (PT)

Pasting temperature (PT) ranged from 62.7 to 83.35 °C. The highest PT was reported in the hybrid cultivar N12 (83.35 °C), and the lowest in the landrace TRZ (62.7 °C). The results of this study are in agreement with other authors [19,22,41,43]. Pasting temperature varied significantly (*p* < 0.05) across the cooking banana cultivars. The pasting temperature of hybrid cultivars varied from 74.65 to 82.35 °C, whereas that of landraces ranged between 62.7 and 81.05 °C. Hybrid cultivars were seen to have a higher pasting temperature compared to landraces. High values of PT are an indication of high resistance to starch granule swelling and consequent rupture [47]. The results from this study showed that landraces were seen to swell and rupture at lower temperatures compared to hybrids. This implies that industries would save a lot of energy if they used these starches in their processes. Pasting temperature is a key parameter used to assess the quality and functionality of starch in food products. It influences the texture of food products. Lower pasting temperatures indicate a higher tendency for starch gelatinization and greater softening during cooking or processing. Therefore, the landrace cultivar TRZ, with its low pasting temperature, may yield a softer and more palatable banana texture compared to the other cultivars. Banana cultivars with low pasting temperatures, such as TRZ, may be more suitable for applications that require rapid starch gelatinization, such as banana-based batters, purees, or desserts.

On the other hand, hybrids with higher pasting temperatures, such as the hybrid N12, might be preferred for applications where starch retrogradation is desired, such as the production of starchy banana snacks or flour. Breeders can use pasting temperature as a selection criterion to develop new hybrids with specific textural and processing characteristics. For example, if the aim is to create a new hybrid with a softer texture, breeders can prioritize crosses involving parents that exhibit lower pasting temperatures. The variation in pasting temperature allows breeders to create hybrids with desired textural and processing characteristics, enabling the development of new banana varieties that cater to diverse consumer preferences and application requirements in the food industry.

## 4. Conclusions

Regarding the functional properties, noteworthy variations were observed among the samples. The swelling power, water absorption index (WAI), and water absorption capacity (WAC) exhibited substantial differences, indicating diverse capabilities for moisture retention and absorption. Notably, samples N7, N8, and N11 demonstrated particularly high swelling power and water absorption properties, suggesting their potential applicability in moisture-sensitive industries.

In terms of pasting properties, the samples exhibited distinct rheological behaviors. Peak viscosity, trough viscosity, breakdown viscosity, final viscosity, and setback viscosity provided crucial information about the thickening and retrogradation tendencies of the samples. Samples N6 and MSK displayed notably high peak viscosities, indicative of their potential use in applications where viscosity is a critical parameter.

Additionally, the pasting time and temperature are essential factors for process optimization. Samples with shorter pasting times, such as N2 and KBZ, may be advantageous in industries where efficiency and speed are paramount. Similarly, those with lower pasting temperatures, such as N7 and TRZ, may find applications in processes where heat sensitivity is a concern.

Generally, the diverse functional and pasting properties exhibited by the samples highlight their potential for various industrial applications. The choice of sample should be made based on specific requirements, such as moisture retention, viscosity, and process conditions. Further research and development efforts could focus on harnessing the unique properties of these samples to enhance their applicability in specific industries.

## Figures and Tables

**Figure 1 foods-12-04323-f001:**
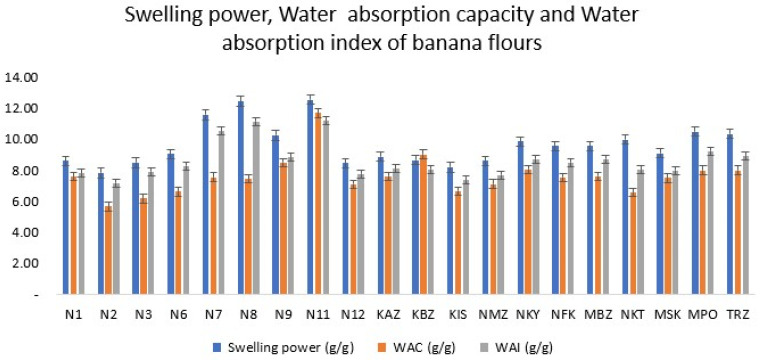
Graphical representation of the swelling power, water absorption capacity, and water absorption index of the banana flours.

**Figure 2 foods-12-04323-f002:**
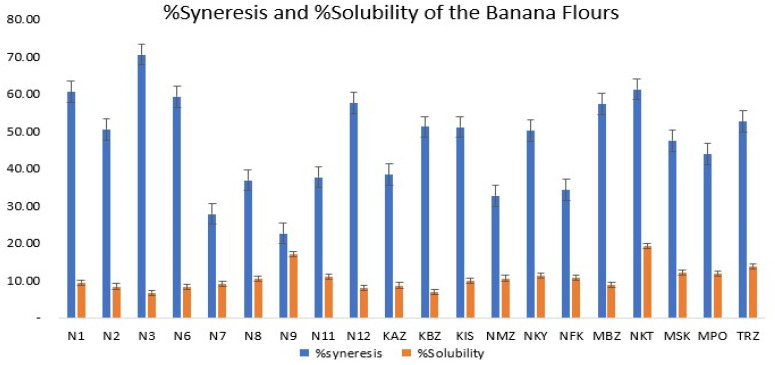
Graphical representation of the freeze–thaw stability (%syneresis) and %solubility.

**Table 1 foods-12-04323-t001:** Cooking banana genotypes used for this study.

Cultivar	Cultivar Code	Clone Set	Cultivar Type	Date and Year of Harvest
Kazirakwe	KAZ	Nakabululu	Landrace	3 November 2020
Kibuzi	KBZ	Nakitembe	Landrace	4 October 2022
Kisansa	KIS	Musakala	Landrace	16 December 2020
Namwezi	NMZ	Nfuuka	Landrace	25 February 2021
Nakinyika	NKY	Nfuuka	Landrace	13 September 2019
Nakitembe	NKT	Nakitembe	Landrace	16 February 2020
Mpologoma	MPO	Musakala	Landrace	13 September 2019
Nfuuka	NFK	Nfuuka	Landrace	18 February 2021
Tereza	TRZ	Nfuuka	Landrace	13 September 2019
Mbwazirume	MBZ	Nakitembe	Landrace	15 March 2022
Musakala	MSK	Musakala	Landrace	25 March 2022
NARITA 1	N1	NA	Hybrid	1 November 2022
NARITA 2	N2	NA	Hybrid	26 November 2018
NARITA 3	N3	NA	Hybrid	4 January 2017
NARITA 6	N6	NA	Hybrid	15 April 2019
NARITA 7	N7	NA	Hybrid	1 October 2019
NARITA 8	N8	NA	Hybrid	10 February 2021
NARITA 9	N9	NA	Hybrid	8 May 2018
NARITA 11	N11	NA	Hybrid	31 May 2022
NARITA 12	N12	NA	Hybrid	18 October 2018

NA = Does not belong to clone sets.

**Table 2 foods-12-04323-t002:** Functional properties of cooking banana flour.

Sample Name	Sample Code	Swelling Power (g/g)	WAI (g/g)	WAC (g/g)	%Syneresis	%Solubility
NARITA1	N1	8.85 ± 0.26	7.80 ± 0.21	7.58 ± 0.84	60.55 ± 3.85	9.45 ± 0.36
NARITA2	N2	7.83 ± 0.17	7.16 ± 0.17	5.66 ± 0.02	50.49 ± 2.12	8.39 ± 0.65
NARITA3	N3	8.64 ± 0.11	7.92 ± 0.11	6.18 ± 0.11	70.53 ± 1.25	6.66 ± 0.15
NARITA6	N6	9.42 ± 0.09	8.29 ± 0.09	6.62 ± 2.05	59.27 ± 1.19	8.24 ± 0.14
NARITA7	N7	9.99 ± 1.51	10.51 ± 1.51	7.54 ± 0.83	27.83 ± 0.18	9.04 ± 1.04
NARITA8	N8	12.02 ± 0.51	11.13 ± 0.51	7.44 ± 0.82	36.85 ± 1.01	10.49 ± 0.21
NARITA9	N9	8.56 ± 1.62	8.84 ± 1.62	8.48 ± 1.41	22.59 ± 1.01	13.16 ± 3.76
NARITA11	N11	11.94 ± 0.56	11.16 ± 0.56	11.68 ± 0.4	37.66 ± 0.13	11.04 ± 0.84
NARITA12	N12	7.95 ± 0.46	7.78 ± 0.36	7.09 ± 1.34	57.54 ± 1.21	7.99 ± 0.83
KAZIRAKWE	KAZ	8.66 ± 0.23	8.11 ± 0.40	7.61 ± 0.83	38.38 ± 0.19	8.67 ± 0.93
KIBUZI	KBZ	8.75 ± 0.46	8.05 ± 0.43	9.01 ± 1.65	51.19 ± 3.59	6.93 ± 0.09
KISANSA	KIS	9.34 ± 1.01	7.37 ± 0.78	6.64 ± 0.83	51.10 ± 4.71	9.88 ± 1.50
MBWAZIRUME	MBZ	9.27 ± 0.28	8.71 ± 0.20	7.60 ± 0.82	57.30 ± 8.56	8.71 ± 0.66
MPOLOGOMA	MPO	10.68 ± 0.46	9.24 ± 0.23	7.99 ± 0.77	37.66 ± 11.11	11.73 ± 2.47
MUSAKALA	MSK	9.32 ± 0.17	7.98 ± 0.16	7.49 ± 0.01	47.51 ± 2.69	12.02 ± 0.12
NFUUKA	NFK	9.40 ± 0.99	8.51 ± 0.79	7.54 ± 0.79	34.24 ± 2.37	10.82 ± 0.09
NAKITEMBE	NKT	10.83 ± 0.53	8.04 ± 0.50	6.55 ± 0.06	61.23 ± 15.04	19.14 ± 0.52
NAKINYIKA	NKY	9.97 ± 0.75	8.71 ± 0.29	8.05 ± 1.09	50.18 ± 12.85	11.19 ± 3.43
NAMWEZI	NMZ	9.12 ± 0.46	7.69 ± 0.41	7.10 ± 1.41	32.69 ± 0.83	10.58 ± 0.09
TEREZA	TRZ	10.65 ± 0.71	8.90 ± 0.51	7.99 ± 0.84	52.60 ± 0.93	13.73 ± 1.90

All data represent means of triplicates ± standard deviations.

**Table 3 foods-12-04323-t003:** Pasting properties of flour from different cooking banana cultivars.

Codes	Type	Peak Vis.	Trough Vis.	Breakdown Vis.	Final Vis.	Setback Vis.	Peak Time	Pasting Temp
N1	Hybrid	2835 ± 454.60	1598 ± 126.40	1235 ± 328.70	3690 ± 113.30	2092 ± 91.80	5.70 ± 0.30	79.69 ± 1.40
N2	Hybrid	1451 ± 174.30	949 ± 102.60	883 ± 78.00	1355 ± 146.40	786 ± 44.10	4.65 ± 0.00	78.84 ± 0.40
N3	Hybrid	2855 ± 364.20	1575 ± 171.10	1280 ± 205.80	3842 ± 431.3	2267 ± 275.4	5.50 ± 0.20	78.83 ± 0.40
N6	Hybrid	3608 ± 562.90	1869 ± 293.40	2120 ± 269.40	3685 ± 301.20	2197 ± 7.80	4.93 ± 0.10	79.98 ± 0.00
N7	Hybrid	3505 ± 269.40	1775 ± 100.90	2110 ± 201.2	3497 ± 191.50	2102 ± 170.80	4.82 ± 0.10	74.65 ± 2.80
N8	Hybrid	2193 ± 122.90	1443 ± 400.20	1130 ± 722.7	2983 ± 1338.10	1920 ± 987.80	5.47 ± 0.20	81.08 ± 0.50
N9	Hybrid	3281 ± 344.40	2140 ± 103.20	1141 ± 241.1	3362 ± 159.10	1222 ± 55.90	5.0 ± 0.10	79.85 ± 0.10
N11	Hybrid	1697 ± 552.10	1193 ± 58.90	884 ± 493.5	2030 ± 513.70	1217 ± 455.00	5.27 ± 0.06	81.25 ± 3.70
N12	Hybrid	2939 ± 192.00	1884 ± 134.50	1054 ± 103.5	3823 ± 200.80	1938 ± 91.80	5.68 ± 0.00	82.35 ± 0.00
KAZ	Landrc	2363 ± 227.00	1367 ± 54.40	1376 ± 172.5	3060 ± 146.40	2073 ± 91.90	5.2 ± 0.20	77.43 ± 0.10
KBZ	Landrc	1535 ± 14.80	976 ± 12.70	939 ± 2.10	1359 ± 9.90	763 ± 2.80	4.7 ± 0.00	77.85 ± 0.60
KIS	Landrc	457 ± 82.70	423 ± 72.80	414 ± 9.90	567 ± 97.60	525 ± 24.70	6.0 ± 0.20	81.05 ± 0.60
NMZ	Landrc	3155 ± 81.30	1904 ± 169.70	1251 ± 88.40	3637 ± 84.10	1733 ± 85.60	5.77 ± 0.10	79.90 ± 0.10
NKYK	Landrc	3980 ± 194.10	2081 ± 129.00	1949 ± 110.3	3511 ± 145.20	1483 ± 61.20	4.82 ± 0.00	77.68 ± 0.60
NFK	Landrc	3959 ± 142.10	1943 ± 36.10	2396 ± 106.1	4111 ± 48.10	2549 ± 12.00	5.10 ± 0.00	77.40 ± 0.10
MBZ	Landrc	2377 ± 360.10	1422 ± 99.80	1153 ± 522.80	2618 ± 172.50	1361 ± 317.50	5.48 ± 0.90	78.17 ± 1.80
NKT	Landrc	2703 ± 627.90	1606 ± 161.90	1478 ± 466.00	3034 ± 546.60	1808 ± 384.70	5.13 ± 0.20	77.85 ± 0.60
MSK	Landrc	6265 ± 52.30	4254 ± 67.90	1872 ± 72.80	5004 ± 24.00	809 ± 77.10	5.51 ± 0.40	78.16 ± 0.00
MPO	Landrc	2905 ± 50.20	1407 ± 178.40	1498 ± 176.40	3196 ± 130.30	1789 ± 303.00	5.02 ± 0.10	78.03 ± 0.50
TRZ	Landrc	2930 ± 517.90	1733 ± 309.40	1197 ± 231.30	3679 ± 139.00	1946 ± 320.60	5.58 ± 0.20	62.70 ± 13.50

Analysis was conducted in triplicates. Landrc = Landrace. These are values of Means ± Standard deviations.

**Table 4 foods-12-04323-t004:** Pearson’s correlation coefficients of functional and pasting characteristics.

	Peak Vis.	Trough Vis.	Breakdown Vis.	Final Vis.	Setback Vis.	Peak Time	Pasting T.	Swelling Power	%Solubility	WAC (g/g)	WAI (g/g)	%Syneresis
Peak Vis.	1	0.95	0.78	0.90	0.29	−0.08	−0.16	0.05	0.22	−0.06	0.00	−0.11
Trough Vis.	0.95	1	0.57	0.82	0.08	0.05	−0.09	0.03	0.26	−0.02	−0.03	−0.11
BDV	0.78	0.57	1	0.71	0.57	−0.40	−0.19	0.16	0.08	−0.11	0.14	−0.16
Final Vis.	0.90	0.82	0.71	1	0.62	0.12	−0.20	0.09	0.18	−0.11	0.05	−0.03
Setback	0.29	0.08	0.57	0.62	1	0.04	−0.19	0.18	−0.05	−0.17	0.20	0.04
Peak Time	−0.08	0.05	−0.40	0.12	0.04	1	0.07	−0.16	−0.02	−0.12	−0.17	0.22
Pasting T.	−0.16	−0.09	−0.19	−0.20	−0.19	0.07	1	−0.14	−0.23	−0.03	−0.08	−0.09
Swelling P.	0.05	0.03	0.16	0.09	0.18	−0.16	−0.14	1	0.35	0.60	0.97	−0.49
%Solubility	0.22	0.26	0.08	0.18	−0.05	−0.02	−0.23	0.35	1	0.13	0.13	−0.25
WAC (g/g)	−0.06	−0.02	−0.11	−0.11	−0.17	−0.12	−0.03	0.60	0.13	1	0.62	−0.40
WAI (g/g)	0.00	−0.03	0.14	0.05	0.20	−0.17	−0.08	0.97	0.13	0.62	1	−0.49
%Syneresis	−0.11	−0.11	−0.16	−0.03	0.04	0.22	−0.02	−0.49	−0.25	−0.40	−0.49	1

## Data Availability

All data for this study are included in the manuscript.

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
