# Peer review of "Internal Quality Assessment of East African Highland Cooking Banana (Musa spp.) Flour: Significance for Breeding and Industrial Applications"

_foods, 2023, doi:10.3390/foods12234323_

Round 1

Reviewer 1 Report

Introduction

Lines: 23-24 I recommend not to mention the possibility of using it as infant formulation, as those products have a lot of other requirements and this paper is out of this topic.

Line 45: “eating quality” is not an official indicator, name it differently f.ex. sensory characteristics.

Line 53-54: “Screening based on only resistance and agronomic traits has resulted in low uptake of improved bananas for food” this sentence is not clear. Resistance to what? “Improved bananas for food”? for nutritional purposes? 

In the introduction the cooking bananas should be better characterized. What is it? How it is use? Etc.

Materials and Method

Lines 104-105 “fingers” is not a commonly used term for the pulp of banana. 65oC is rather too low for drying. In the cited methodology a temperature 100 oC is recommended.

Additionally, I suggest to present some of the data as charts to make it easier to interpretate and more interesting.

Conclusions could be more specific. Not only a general statement but more comparison and pointing the best hybrids and landrace cultivars. 

The article describes only the basic physical properties, therefore, it should be more professionally written. 

Reviewer 2 Report

Although the paper is interesting and could be very important for the potential industrial applications of banana powder, the data presented in the manuscript shows that some experiment were not done correctly (SD exceed the mean values of the measurement), which, as a consequence, questions the liability and the validity of the results and the conclusions presented, and cannot, therefore, be accepted as a valid scientific paper with sound conclusions. Please see the comments below.

The title and the abstract of the paper should be modified to clearly state that the banana POWDERS were analyzed, not the fresh bananas. Suggestion: "Internal Quality Assessment of East African Highland Cooking
Banana (Musa spp.) powders: Significance for Breeding and Industrial
Applications." The same is valid for the first sentence in the Abstract - please modify as follows: "This study assessed the internal quality traits of East African Highland cooking banana powders, exploring their significance for breeding and potential industrial applications.

Keywords should also be modified to contain "cooking banana powder"

P1, L14: Please correct "Nine" to "nine"

P1, L37: correct "bananas" to "banana"

P2, L60 - 71: This subsection does not make sense. The topic of your study are banana powders, yet you describe research made on cereals in this subsection. Please remove this subsection or rewrite it so that it deals with banana flour properties.

P2, subsection Materials: Please indicate the year and the month the bananas were harvested.

Please numerate the equations.

P5, subsection Statistical analysis: Did you test the normality of data distribution prior to performing ANOVA? Only if the data was normally distributed, you can use parametric statistical tests such as ANOVA.

P6, Table 2. I am concerned about the data on WAC presented in Table 2. The results for N12, N1, N3, N2, KBZ, KAZ, NKYK, NMZ, MBZ, NFK and KIS are not experimentally valid, nor suitable for further statistical analysis, discussion and conclusions, since the values of SD are more than half the value of the measured WAC, or, in some cases SDs even exceed the values of WAC. This is a clear indicator that the experimental analysis was not conducted correctly and the results obtained by that analysis are invalid. Either remove the data on WAC from the Table, discussion and conclusions altogether, or, instead of 2 parallel measurements, make at least 5 on the same sample and then review the results and decide whether they are suitable for publication. The same goes for solubility data (N12, N1, KBZ, NFK).

P7, L244 - 261 : The discussion is not valid due to the invalid experimental results.

P7, L277 - 295 : The discussion is not valid due to the invalid experimental results.

P10, Table 3: There are no SDs (or any other type of statistical errors) presented in Table 3.

P12, Table 4. The statistical analysis is questionable, due to the results with high SD values presented in Table 2 (WAC and solubility, see previous comments).

The conclusions are also questionable due to some invalid experimental results.

Moderate editing required.

Reviewer 3 Report

The article addresses an important issue for Uganda, which is the cultivation of bananas for consumption.

The article is properly structured, but it needs some adjustments in the conduct of the discussion. 

The authors use the phrase "starch solution, starch suspension." This is inappropriate and suggests that the starch was isolated and tested during the research experiment, while this was not the case. This term should be modified and replaced with "banana flour". 

The article is also missing 3 important determinations - a comparison of the color of the raw banana, the color of the flour, and the color of the color glue obtained from the RVA. The color scale in the CIELab system and the change relative to the subsequent processing step, together with an analysis of the content of simple sugars (there may be a method with DNS) will make it possible to indicate in which cases the viscosity of the glue from RVA depended on the starch system, and in which cases the level of simple sugars was also responsible. 

One more minor note to the Introduction - please replace the phrase "mother earth" with "planet earth".

The second remark is more serious - in the description of the RVA method, it was indicated that the total weight was 28 g, while the description shows that 2.5 g of sample and 25 ml of water were taken, which does not give a total of 28 g. In addition, graphs of pasting curves would be very useful.

Round 2

Reviewer 1 Report

Thank you for addressing my comments. In my opinion the article is now suitable to be published in Foods journal.

Reviewer 2 Report

The Authors have answered all of my questions successfully.

Moderate English editing required.